# Impact of Institutional Monthly Volume of Transcatheter Edge-to-Edge Repair Procedures for Significant Mitral Regurgitation: Evidence from the GIOTTO-VAT Study

**DOI:** 10.3390/medicina61050904

**Published:** 2025-05-16

**Authors:** Nicola Corcione, Paolo Ferraro, Filippo Finizio, Michele Cimmino, Michele Albanese, Alberto Morello, Giuseppe Biondi-Zoccai, Paolo Denti, Antonio Popolo Rubbio, Francesco Bedogni, Antonio L. Bartorelli, Annalisa Mongiardo, Salvatore Giordano, Francesco De Felice, Marianna Adamo, Matteo Montorfano, Francesco Maisano, Giuseppe Tarantini, Francesco Giannini, Federico Ronco, Emmanuel Villa, Maurizio Ferrario, Luigi Fiocca, Fausto Castriota, Angelo Squeri, Martino Pepe, Corrado Tamburino, Arturo Giordano

**Affiliations:** 1Unità Operativa di Interventistica Cardiovascolare, Pineta Grande Hospital, 81030 Castel Volturno, Italy; nicolacorcione72@gmail.com (N.C.); filippo.finizio@pinetagrande.it (F.F.); cimmino.michele88@gmail.com (M.C.); albogottlieb83@gmail.com (A.M.); arturgiordano@gmail.com (A.G.); 2Unità Operativa di Emodinamica, Santa Lucia Hospital, 80047 San Giuseppe Vesuviano, Italy; ferpa961@gmail.com (P.F.); mikelealbanese@gmail.com (M.A.); 3Department of Medico-Surgical Sciences and Biotechnologies, Sapienza University of Rome, 04100 Latina, Italy; 4Maria Cecilia Hospital, GVM Care & Research, 48033 Cotignola, Italy; 5Department of Cardiac Surgery, Vita-Salute San Raffaele University, IRCCS San Raffaele Scientific Institute, 20132 Milan, Italy; paolodenti@hotmail.com; 6Department of Cardiology, IRCCS Policlinico San Donato, San Donato Milanese, 20097 Milan, Italy; antoniorubbio@yahoo.it (A.P.R.); francesco.bedogni@grupposandonato.it (F.B.); 7IRCCS Ospedale Galeazzi-Sant’Ambrogio, 20157 Milan, Italy; antonio.bartorelli@grupposandonato.it; 8Department of Biomedical and Clinical Sciences, University of Milan, 20133 Milan, Italy; 9Division of Cardiology, Department of Medical and Surgical Sciences, “Magna Graecia” University, 88100 Catanzaro, Italy; amongiardo@tin.it (A.M.); sasigiordano@gmail.com (S.G.); 10Division of Interventional Cardiology, Azienda Ospedaliera S. Camillo Forlanini, 00152 Rome, Italy; f.defelice1966@gmail.com; 11Cardiac Catheterization Laboratory and Cardiology, ASST Spedali Civili di Brescia, 25121 Brescia, Italy; mariannaadamo@hotmail.com; 12Department of Medical and Surgical Specialties, Radiological Sciences, and Public Health, University of Brescia, 25123 Brescia, Italy; 13School of Medicine, Vita-Salute San Raffaele University, 20132 Milan, Italy; montorfano.matteo@hsr.it (M.M.); maisano.francesco@hsr.it (F.M.); 14Interventional Cardiology Unit, IRCCS San Raffaele Scientific Institute, 20132 Milan, Italy; giuseppe.tarantini.1@gmail.com; 15Interventional Cardiology Unit, Department of Cardiac, Thoracic and Vascular Science, University of Padua, 35131 Padua, Italy; 16Division of Cardiology, IRCCS Ospedale Galeazzi-Sant’Ambrogio, 20157 Milan, Italy; giannini_fra@yahoo.it; 17Interventional Cardiology, Department of Cardio-Thoracic and Vascular Sciences, Ospedale dell’Angelo, AULSS3 Serenissima, Mestre, 30174 Venezia, Italy; federicoronco@yahoo.it; 18Cardiac Surgery Unit and Valve Center, Poliambulanza Foundation Hospital, 25124 Brescia, Italy; emmanuel.villa@poliambulanza.it; 19Division of Cardiology, Fondazione IRCCS Policlinico S. Matteo, 27100 Pavia, Italy; m.ferrario@smatteo.pv.it; 20Cardiovascular Department, Papa Giovanni XXIII Hospital, 24129 Bergamo, Italy; luigifiocca@gmail.com; 21Interventional Cardiology Unit, Maria Cecilia Hospital, GVM Care & Research, 48033 Cotignola, Italy; fcastriota@msn.com (F.C.); asqueri@gvmnet.it (A.S.); 22Division of Cardiology, Department of Interdisciplinary Medicine (D.I.M.), University of Bari Aldo Moro, 70121 Bari, Italy; drmartinopepe@gmail.com; 23Division of Cardiology, Centro Alte Specialità e Trapianti (CAST), Azienda Ospedaliero-Universitaria Policlinico-Vittorio Emanuele, University of Catania, 95123 Catania, Italy; tamburinocorrado@gmail.com

**Keywords:** caseload, experience, mitral regurgitation, MitraClip, volume

## Abstract

*Background and Objectives*: Mitral valve transcatheter edge-to-edge repair (TEER) is a widely adopted therapeutic approach for managing significant mitral regurgitation (MR) in high-risk surgical candidates. While procedural safety and efficacy have been demonstrated, the impact of institutional expertise on outcomes remains unclear. We aimed at evaluating whether the institutional monthly volume of TEER influences short- and long-term clinical results. *Materials and Methods*: This analysis from the multicenter, prospective GIOTTO trial study evaluated the impact of institutional monthly volume on outcomes of TEER to remedy significant mitral regurgitation. Centers were stratified into tertiles based on monthly volumes (≤2.0 cases/month, 2.1–3.5 cases/month, >3.5 cases/month), and key clinical, echocardiographic, and procedural outcomes were analyzed. Statistical analysis was based on standard bivariate tests as well as unadjusted and multivariable adjusted Cox models. *Results*: A total of 2213 patients were included, stratified into tertiles based on institutional procedural volume: 645 (29.1%) patients in the first tertile, 947 (42.8%) patients in the second tertile, and 621 (28.1%) patients in the third tertile. Several baseline differences were found, with some features disfavoring less busy centers (e.g., functional class and surgical risk, both *p* < 0.05), and others suggesting a worse risk profile in those treated in busier institutions (e.g., frailty and history of prior mitral valve intervention, both *p* < 0.05). Procedural success rates were higher in busier centers (*p* < 0.001), and hospital stay was also shorter there (*p* < 0.001). Long-term follow-up (median 14 months) suggested worse outcomes in patients treated in less busy centers at unadjusted analysis (e.g., *p* = 0.018 for death, *p* = 0.015 for cardiac death, *p* = 0.014 for death or hospitalization for heart failure, *p* < 0.001 for cardiac death or hospitalization for heart failure), even if these associations proved no longer significant after multivariable adjustment, except for cardiac death or hospitalization for heart failure, which appeared significantly less common in the busiest centers (*p* < 0.05). Similar trends were observed when focusing on tertiles of overall center volume and when comparing for each center the first 50 cases with the following ones. *Conclusions*: High institutional monthly volume of TEER mitral valve repair appears to correlate with an improved procedural success rate and shorter hospitalizations. Similarly favorable results were found for long-term rates of cardiac death or hospitalization for heart failure. These findings inform on the importance of operator experience and center expertise in achieving state-of-the-art results with TEER, while confirming the usefulness of the proctoring approach when naïve centers begin a TEER program.

## 1. Introduction

Transcatheter edge-to-edge repair (TEER) has emerged as an effective alternative to surgical valve repair in high-risk patients, and to conservative medical therapy only in carefully selected individuals considered unfit for surgery [1,2]. Indeed, given its ability to improve functional outcomes and reduce heart failure hospitalizations, TEER is increasingly performed worldwide [3]. However, the early and long-term effectiveness of TEER is impacted by several factors, ranging from patient to anatomic and procedural ones [4,5,6,7,8,9,10,11]. 

On top of patient, procedural, and operator features, institutional characteristics have been the focus of attentive recommendations and analysis [12]. Indeed, in most settings, TEER is only provided as long as surgical mitral valve repair/replacement is also available. Yet, this minimum requirement might be too lenient because expertise in patient screening and selection, procedural efficiency, and seamless quality of care are paramount to achieve optimal outcomes with TEER [13]. In particular, a number of cutoffs have been proposed and tested formally in the past, under the key premise that higher-volume centers may provide better results, shortly after the procedure as well as subsequently. However, results have been inconsistent so far, with some studies suggesting that no evident cutoff can be envisioned, and others ending up recommending a minimum yearly volume ranging from 8 to 24 cases [14,15,16].

We hypothesized that the institutional monthly volume of TEER could influence short- and long-term clinical results of this technically demanding procedure, in the sense that institutions exhibiting higher monthly volumes could provide better outcomes in comparison to centers with a lower monthly caseload. We thus aimed at appraising the impact of the institutional monthly volume of TEER on early and long-term outcomes, and in order to test this hypothesis, we chose to leverage the extensive and detailed dataset of the ongoing prospective GIOTTO (GIse registry Of Transcatheter treatment of mitral valve regurgitaTiOn) registry, an Italian multicenter observational study including patients undergoing TEER with a MitraClip (Abbott Vascular, Santa Clara, CA, USA) [5]. Indeed, key strengths of GIOTTO include its contemporary stance and follow-up that goes well beyond discharge.

## 2. Methods

This study was based on the dataset accrued in the GIOTTO trial, which is sponsored by the Italian Society of Invasive Cardiology (GISE—Società Italiana di Cardiologia Interventistica, Milan, Italy) and is registered online at ClinicalTrials.gov (NCT03521921) [5]. Notably, ethical approval was obtained from all participating institutions, and all patients provided written informed consent.

For the purpose of this analysis, which we labelled GIOTTO-VAT (volume and time), we mainly focused on comparing tertiles of cases per month, center-wise, with the first tertile up to 2.0 cases per month, the second tertile with more than 2.0 and up to 3.5 cases per month, and the third tertile with more than 3.5 cases per month. Exploratory analyses were conducted according to tertiles of total volume, center-wise, with the first tertile up to 100 cases, the second tertile with more than 100 and up to 200 cases, and the third tertile with more than 200 cases, as well as distinguishing between the first 50 cases per center and the subsequent ones.

Details on baseline variables were collected, including demographic data, comorbidities, functional class, prior cardiac procedures, and medication history. Echocardiographic parameters being assessed included left atrial diameter, left ventricular dimensions, mitral valve gradient, and severity of tricuspid regurgitation. Procedural variables of interest included number and generation of MitraClip devices implanted, fluoroscopy time, device time, and procedural success. Fatal and non-fatal outcomes occurring during the index hospitalization and during follow-up were systematically collected, with specific attention to the following events: death, cardiac death, the composite of death or hospitalization, and the composite of death or hospitalization for heart failure.

Descriptive statistics were computed for all variables, with medians and first and third quartiles provided for continuous variables, and counts and percentages for categorical variables. Bivariate analysis was based on Kruskal–Wallis tests for continuous variables and Fisher exact tests for categorical variables. Censored outcomes were analyzed with Cox proportional hazard models, unadjusted as well as adjusted for potential confounders. Notably, the following potential confounders were forced into the adjusted models: age, gender, smoking history, dyslipidemia, degenerative etiology, baseline functional class, prior mitral valve repair, prior stroke, frailty, peripheral artery disease, surgical risk scores, left atrial diameters, left ventricular diameters, left ventricular volumes, tenting area, severe calcification, prolapse, severity of tricuspid regurgitation, concomitant ECG abnormalities, atrial fibrillation, and significant coronary artery disease. No missing data imputation was performed. Statistical significance was set at a 2-tailed *p*-value of 0.05, without multiplicity adjustments, and all analyses were conducted using Stata 18 (StataCorp, College Station, TX, USA).

## 3. Results

A total of 2213 patients were included, with 645 (29.1%) individuals treated in centers performing ≤ 2.0 cases/month, 947 (42.8%) treated in institutions reporting between 2.1 and 3.5 cases/month, and 621 (28.1%) patients treated in hospitals with >3.5 cases/month (Table 1). Several differences according to such stratifications were found in key baseline features, with some suggesting a higher complexity in patients treated in lower-volume centers, such as age, functional class, and surgical risk score (all *p* < 0.05), and others suggesting a higher risk in those treated in busier centers, such as smoking, dyslipidemia, prior mitral valve intervention, prior stroke, peripheral artery disease, and frailty score (all *p* < 0.05).

Other significant differences were found for left ventricular dimensions and function, and mitral valve tenting area (Table 2), disfavoring less busy centers (all *p* < 0.05), and for left atrial dimensions, mitral valve calcification, mitral valve prolapse, tricuspid regurgitation severity, concomitant ECG abnormalities, and prevalence of atrial fibrillation, disfavoring higher-volume institutions (all *p* < 0.05).

In terms of procedural details, significant differences were found in rates of implantation of multiple MitraClips, type of MitraClip used, device time, and fluoroscopy time (all *p* < 0.05; Table 3). Notably, device success rates were similar across tertiles, but procedural success was marginally albeit significantly higher in busier centers (*p* < 0.001), with concomitantly lower rates of severe residual mitral regurgitation (*p* = 0.001). Patients in high-volume centers experienced fewer in-hospital bleeding events (*p* = 0.006) but more vascular complications (*p* = 0.008), without significant differences in in-hospital mortality (*p* = 0.123). Length of hospital stay was significantly shorter in higher-volume centers (*p* < 0.001).

During a median follow-up of 14 months, details on a total of 539 (24.4%) deaths, 286 (12.9%) cardiac deaths, 685 (31.0%) deaths or hospitalizations, and 359 (16.2%) cardiac deaths or hospitalizations for heart failure were accrued (Table 4). Moderate or severe mitral regurgitation was reported overall in 308 (14.3%). Survival analysis was performed using first unadjusted models (Table 5), and these suggested worse outcomes in less busy centers for death, cardiac death, the composite of death or hospitalization, and the composite of cardiac death or hospitalization for heart failure (all *p* < 0.05). However, after taking into account potential confounders, all these differences were no longer significant, except for cardiac death or hospitalization for heart failure, which appeared significantly less common in the busiest centers (all *p* < 0.05; Figure 1).

Similar trends were observed when leveraging tertiles of overall center volume and when comparing for each center the first 50 cases with the following ones (Appendix A), with higher procedural success rates in busier/more experienced centers, and similarly trends favoring them for long-term clinical outcomes.

## 4. Discussion

A high institutional monthly volume of TEER mitral valve repair appears to correlate with improved procedural success rate and shorter hospitalizations. Similarly favorable results were found for long-term rates of cardiac death or hospitalization for heart failure.

The evidence base appraising the impact of institutional volume and expertise on TEER outcomes is complex and heterogeneous, but some studies have indeed suggested that some thresholds are important to achieve satisfactory procedural, in-hospital, and mid-term outcomes (Appendix A) [10,12,14,15,16,17]. The findings from the present GIOTTO-VAT study support the concept that institutional volume in TEER may significantly impact on clinical outcomes. Notably, centers with higher monthly case volumes demonstrated greater procedural success. Furthermore, similarly favorable results were evident for long-term outcomes at unadjusted and adjusted analysis. These results align with existing evidence in the field of structural heart interventions, where operator and institutional experience have been shown to positively impact clinical outcomes [10,11].

Interestingly, while higher-volume centers showed better procedural metrics, long-term clinical outcomes, including mortality, were comparable across tertiles. This suggests that while experience improves technical execution and immediate safety, patient selection and underlying clinical conditions remain key determinants of long-term prognosis. Moreover, the impact of experienced proctors affiliated with high-volume centers providing careful guidance to lower-volume institutions cannot be discounted, in person or remotely. Indeed, the impact of proctors should be carefully appraised in future studies, together with detailed analyses on post-procedural care and management protocols.

A notable finding was the reduced hospitalization duration in high-volume centers compared to lower-volume ones. This may reflect more efficient perioperative management, shorter procedural times, and faster post-operative recovery [18]. Shorter hospital stays can also reduce healthcare costs, which is a significant consideration in the growing adoption of TEER for mitral regurgitation management, thus reinforcing the call for established expertise in all phases of management of patients with significant mitral regurgitation [19]. 

Our results prove informatively complementary to those reported by Grayburn and colleagues, who analyzed the American College of Cardiology/Society of Thoracic Surgeons Transcatheter Valve Therapy Registry and the Society of Thoracic Surgeons Adult Cardiac Surgery Database comparing institutions with high volumes for mitral valve repair, including surgical interventions [20]. In this study of 41,834 patients, the TEER success rate was similar in low- vs. high-volume centers, but 1-year survival and freedom from heart failure readmission were better in higher-volume hospitals.

This study has several key limitations that should be carefully considered by readers. First, despite some clear hints that higher volumes over time are associated with better outcomes in patients undergoing TEER for significant mitral regurgitation, results were not altogether consistent [21]. In addition, even adjusted analyses cannot be considered devoid of risk of residual confounding, such as ethnic features, which were not collected in our study. This is a key drawback, as minorities often have a more difficult time in accessing high-volume centers [22]. Furthermore, the risk of duplicity and type I error inflation due to a plethora of statistical tests remains substantial, and thus external replication of the present findings is paramount [23].

## 5. Conclusions

The present GIOTTO-VAT study suggests that hospitals with a higher volume of cases over time may yield improved procedural success rates, with ensuing shorter hospitalizations. Long-term rates of cardiac death or hospitalization for heart failure also favored centers with higher case/month figures. These findings inform on the importance of operator experience and center expertise in achieving state-of-the-art results with TEER, while confirming the usefulness of the proctoring approach when naïve centers begin a TEER program.

## Figures and Tables

**Figure 1 medicina-61-00904-f001:**
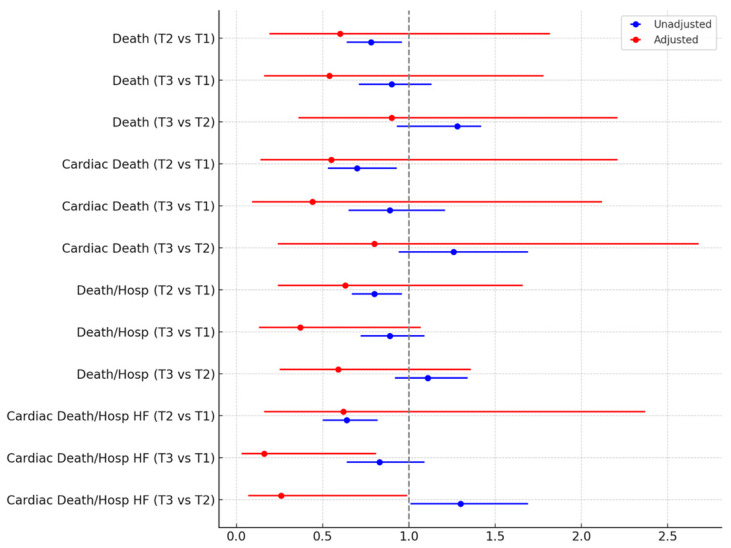
Hazard ratios (95% confidence intervals) at unadjusted and adjusted Cox proportional hazard analysis comparing different tertiles (T) of monthly volume, center-wise. HF = heart failure; Hosp = hospitalization.

**Table 1 medicina-61-00904-t001:** Baseline clinical features according to tertiles of monthly volume, center-wise *.

Feature	First Tertile	Second Tertile	Third Tertile	*p*
Patients	645	947	621	-
Age (years)	78 (72; 83)	77 (70; 82)	79 (70; 83)	<0.001 ^¶,$^
Female gender	211 (32.7%)	586 (61.9%)	380 (61.2%)	0.039
Body mass index	24.9 (22.7; 27.6)	24.8 (22.1; 27.8)	23.4 (2.0; 25.6)	0.356
Smoking history	72 (11.1%)	88 (9.3%)	163 (26.3%)	<0.001
Hypertension	462 (71.6%)	696 (73.5%)	445 (71.7%)	0.628
Dyslipidemia	212 (32.9%)	256 (27.0%)	267 (43.0%)	<0.001
Diabetes mellitus	157 (24.3%)	220 (23.2%)	144 (23.2%)	0.851
Diagnosis				0.002
Degenerative MR	176 (27.3%)	330 (34.9%)	198 (31.9%)	
Functional dilated MR	199 (30.9%)	277 (29.3%)	180 (29.0%)	
Functional ischemic MR	194 (30.1%)	230 (24.3%)	194 (31.2%)	
Mixed etiology	76 (11.8%)	110 (11.6%)	49 (7.9%)	
New York Heart Association class				<0.001
I	4 (0.6%)	11 (1.2%)	17 (2.8%)	
II	110 (17.1%)	215 (22.7%)	196 (31.9%)	
III	452 (70.1%)	660 (69.7%)	350 (57.0%)	
IV	79 (12.3%)	61 (6.4%)	51 (8.3%)	
Coronary artery disease				0.711
None	201 (63.8%)	134 (60.1%)	275 (64.0%)	
Single vessel disease	49 (15.6%)	42 (18.8%)	67 (15.6%)	
Two vessel disease	25 (7.9%)	23 (10.3%)	48 (11.2%)	
Three vessel disease	22 (7.0%)	13 (5.8%)	22 (5.1%)	
Left main disease	18 (5.7%)	11 (4.9%)	18 (4.2%)	
Prior pacemaker implantation				
Prior myocardial infarction	213 (33.0%)	312 (33.0%)	194 (31.2%)	0.735
Prior coronary artery bypass grafting	83 (12.9%)	126 (13.3%)	104 (16.8%)	0.087
Prior mitral valve intervention	17 (2.6%)	14 (1.5%)	38 (6.1%)	<0.001
Prior cerebrovascular event				<0.001
None	601 (93.2%)	891 (94.1%)	548 (88.2%)	
Transient ischemic attack	12 (1.9%)	14 (1.5%)	31 (5.0%)	
Minor stroke	17 (2.6%)	22 (2.3%)	12 (1.9%)	
Major stroke	15 (2.3%)	20 (2.1%)	30 (4.8%)	
Peripheral artery disease	54 (8.4%)	38 (4.0%)	74 (11.9%)	<0.001
Frailty	116 (18.0%)	118 (12.5%)	418 (67.3%)	<0.001
Dialysis	15 (2.3%)	18 (1.9%)	9 (1.5%)	0.521
Logistic EuroSCORE	10.5 (6.5; 16.4)	10.4 (6.2; 19.5)	3.9 (3.4; 4.5)	0.007 ^¶,#^
CHADS2 score	2 (2; 3)	2 (2; 3)	2 (1; 3)	0.004 ^¶,#^
CHADS2Vasc score	4 (3; 5)	4 (3; 5)	4 (3; 4)	0.443

* Descriptive statistics are based on count (%) or median (1st quartile; 3rd quartile), whereas *p* values for inferential statistics are based on Fisher exact or Kruskal–Wallis tests (with ^¶^, ^#^, and ^$^ representing, respectively, *p* < 0.05 at Dunn post hoc tests for tertile 1 vs. tertile 2, tertile 1 vs. tertile 3, and tertile 2 vs. tertile 3); MR = mitral regurgitation.

**Table 2 medicina-61-00904-t002:** Baseline imaging and ECG features according to tertiles of monthly volume, center-wise *.

Feature	First Tertile	Second Tertile	Third Tertile	*p*
Patients	645	947	621	-
LA AP diameter (mm)	45 (40; 50)	49 (44; 55)	49 (45; 55)	0.089
LV EDD (mm)	61 (54; 68)	58 (52; 64)	58 (50; 65)	<0.001 ^¶,#^
LV ESD (mm)	49 (38; 56)	41 (33; 51)	45 (35; 54)	<0.001 ^¶,#,$^
LV EDV (mL)	164 (120; 212)	137 (105; 181)	140 (100; 189)	<0.001 ^¶,#^
LV ESV (mL)	97 (55; 146)	73 (48; 116)	81 (47; 130)	<0.001 ^¶,#,$^
LVEF (%)	38 (30; 55)	42 (31; 55)	40 (30; 57)	0.008 ^¶^
Tenting area (cm^2^)	3.0 (2.2; 3.7)	2.3 (1.6; 3.0)	1.9 (1.3; 2.4)	<0.001 ^¶,#,$^
Mean mitral valve gradient (mm Hg)	2 (1; 3)	2 (1; 2)	2 (2; 3)	0.148
Severe mitral regurgitation	495 (76.7%)	739 (78.0%)	493 (79.4%)	0.525
Severe mitral calcification	22 (3.4%)	28 (3.0%)	55 (8.9%)	<0.001
Mitral valve prolapse	163 (25.3%)	262 (27.7%)	205 (33.0%)	0.007
Flail leaflet	132 (20.5%)	176 (18.6%)	146 (23.5%)	0.063
Tricuspid regurgitation				<0.001
None or trace	23 (3.6%)	62 (6.6%)	22 (3.5%)	
Mild	282 (43.7%)	324 (34.2%)	234 (37.7%)	
Moderate	273 (42.3%)	458 (48.4%)	239 (38.5%)	
Severe	67 (10.4%)	103 (10.9%)	126 (20.3%)	
Systolic pulmonary artery pressure (mm Hg)	46 (40; 55)	45 (37; 55)	45 (35; 55)	0.071
Any ECG abnormality	152 (23.6%)	163 (17.2%)	296 (47.7%)	<0.001
Second-degree atrioventricular block	1 (0.2%)	0	3 (0.5%)	0.039
Third-degree atrioventricular block	1 (0.2%)	4 (0.4%)	2 (0.3%)	0.806
Right bundle branch block	17 (2.6%)	20 (2.1%)	18 (2.9%)	0.577
Left bundle branch block	27 (4.2%)	27 (2.9%)	28 (4.5%)	0.165
Atrial fibrillation	111 (17.2%)	118 (12.5%)	248 (39.9%)	<0.001
Coronary angiography performed	315 (48.8%)	223 (23.6%)	430 (69.2%)	<0.001
Coronary artery disease				0.711
No	201 (63.8%)	134 (60.1%)	275 (64.0%)	
1-vessel disease	49 (15.6%)	42 (18.8%)	67 (15.6%)	
2-vessel disease	25 (7.9%)	23 (10.3%)	48 (11.2%)	
3-vessel disease	22 (7.0%)	13 (5.8%)	22 (5.1%)	
Left main disease	18 (5.7%)	11 (4.9%)	18 (4.2%)	

* Descriptive statistics are based on count (%) or median (1st quartile; 3rd quartile), whereas *p* values for inferential statistics are based on Fisher exact or Kruskal–Wallis tests (with ^¶^, ^#^, and ^$^ representing, respectively, *p* < 0.05 at Dunn post hoc tests for tertile 1 vs. tertile 2, tertile 1 vs. tertile 3, and tertile 2 vs. tertile 3); AP = antero-posterior; EDD = end-diastolic diameter; EDV = end-diastolic volume; ESD = end-systolic diameter; ESV = end-systolic volume; LA = left atrium; LV = left ventricle; LVEF = left ventricular ejection fraction.

**Table 3 medicina-61-00904-t003:** Procedural and in-hospital outcomes according to tertiles of monthly volume, center-wise *.

Outcome	First Tertile	Second Tertile	Third Tertile	*p*
Patients	645	947	621	
Implantation of ≥2 MitraClip devices	421 (65.3%)	508 (53.6%)	377 (60.7%)	<0.001
Implantation on NT device	433 (67.1%)	499 (52.7%)	308 (49.6%)	<0.001
Implantation on NTr device	98 (15.2%)	178 (18.8%)	92 (14.8%)	0.062
Implantation of XTr device	160 (24.8%)	358 (37.8%)	249 (40.1%)	<0.001
Device time (minutes)	2.3 (1.5; 3.5)	2.1 (1.5; 2.7)	3.5 (2.1; 4.4)	<0.001 ^¶,#,$^
Fluoroscopy time (minutes)	0.8 (0.6; 1.7)	1.3 (0.8; 1.9)	0.4 (0.2; 0.8)	<0.001 ^#,$^
Device success	641 (99.4%)	942 (99.5%)	620 (99.8%)	0.513
Procedural success	600 (93.0%)	926 (97.8%)	593 (95.5%)	<0.001
Procedural death	3 (0.5%)	0	2 (03%)	0.104
Mean mitral valve gradient at end of procedure	3 (2; 4)	3 (2; 4)	3 (3; 5)	<0.001 ^¶,#,$^
Mitral regurgitation at end of procedure				<0.001
None	386 (59.8%)	561 (59.2%)	453 (73.0%)	
Mild	209 (32.4%)	348 (36.8%)	147 (23.7%)	
Moderate	34 (5.3%)	26 (2.8%)	12 (1.9%)	
Severe	16 (2.5%)	12 (1.3%)	9 (1.5%)	
Inhospital death	18 (2.8%)	20 (2.1%)	24 (3.9%)	0.123
Inhospital stroke	0	0	0	-
Inhospital bleeding				0.006
None	644 (99.8%)	939 (99.2%)	610 (98.2%)	
Minor	0	5 (0.5%)	7 (1.1%)	
Major	0	1 (0.1%)	4 (0.6%)	
Disabling	1 (0.2%)	2 (0.2%)	0	
Inhospital vascular complication	1 (0.2%)	5 (0.5%)	10 (1.6%)	0.008
Days of hospitalization	8 (5; 12)	5 (4; 8)	5 (4; 8)	<0.001^¶,#,$^
Mitral regurgitation at discharge				0.001
None	347 (55.3%)	503 (54.3%)	388 (65.0%)	
Mild	226 (36.0%)	354 (38.2%)	171 (28.6%)	
Moderate	39 (6.2%)	59 (6.4%)	27 (4.5%)	
Severe	15 (2.4%)	11 (1.2%)	11 (1.0%)	
Systolic pulmonary artery pressure (mm Hg)	40 (32; 46)	40 (35; 50)	40 (30; 45)	<0.001 ^¶,$^

* Descriptive statistics are based on count (%) or median (1st quartile; 3rd quartile), whereas *p* values for inferential statistics are based on Fisher exact or Kruskal–Wallis tests (with ^¶^, ^#^, and ^$^ representing, respectively, *p* < 0.05 at Dunn post hoc tests for tertile 1 vs. tertile 2, tertile 1 vs. tertile 3, and tertile 2 vs. tertile 3).

**Table 4 medicina-61-00904-t004:** Cumulative outcomes at follow-up according to tertiles of monthly volume, center-wise *.

Outcome	First Tertile	Second Tertile	Third Tertile	*p*
Patients	645	947	621	
Follow-up (months)	12 (1; 24)	21 (10; 36)	13 (1; 25)	<0.001 ^¶,$^
Death	152 (23.6%)	251 (26.5%)	136 (21.9%)	0.101
Cardiac death	86 (13.3%)	125 (13.2%)	75 (12.1%)	0.764
Hospitalization	107 (16.6%)	114 (12.0%)	68 (11.0%)	0.007
Hospitalization for heart failure	82 (12.7%)	86 (9.1%)	65 (10.5%)	0.070
Death or hospitalization	199 (30.1%)	307 (32.4%)	179 (28.8%)	0.323
Cardiac death or hospitalization for heart failure	142 (22.0%)	189 (20.0%)	127 (20.5%)	0.596
Mitral valve surgery	4 (0.6%)	8 (0.8%)	13 (2.1%)	0.034
Cerebrovascular accident	12 (1.9%)	13 (1.4%)	9 (1.5%)	0.722
New York Heart Association class				0.006
I	59 (14.0%)	159 (21.3%)	55 (13.3%)	
I	270 (64.0%)	425 (56.9%)	254 (61.2%)	
III	85 (20.1%)	153 (20.5%)	98 (23.6%)	
IV	8 (1.9%)	10 (1.3%)	8 (1.9%)	
Atrial fibrillation	124 (19.2%)	135 (14.3%)	258 (41.6%)	<0.001
End-diastolic diameter (mm)	60 (53; 67)	57 (50; 64)	56 (49; 62)	<0.001 ^¶,$^
End-systolic diameter (mm)	48 (38; 56)	40 (31; 50)	40 (35; 53)	<0.001 ^¶,#,$^
End-diastolic volume (mL)	159 (120; 220)	131 (100; 180)	133 (98; 187)	<0.001 ^¶,#^
End-systolic volume (mL)	90 (56; 138)	73 (45; 115)	85 (45; 130)	0.004 ^¶^
Left ventricular ejection fraction (%)	38 (28; 51)	42 (30; 55)	38 (27; 52)	<0.001 ^¶,$^
Mean mitral valve gradient (mm Hg)	3 (3; 5)	4 (3; 5)	4 (3; 5)	0.023 ^#^
Mitral regurgitation				0.113
None	281 (44.7%)	433 (46.7%)	297 (49.8%)	
Mild	250 (39.8%)	365 (39.4%)	219 (36.7%)	
Moderate	66 (10.5%)	103 (11.1%)	53 (8.9%)	
Severe	32 (5.1%)	26 (2.8%)	28 (4.7%)	
Angiotensin receptor blockers	219 (52.4%)	246 (33.1%)	85 (22.3%)	<0.001
Calcium channel antagonists	34 (8.1%)	77 (10.4%)	41 (10.8%)	0.367
Betablockers	348 (82.7%)	362 (75.4%)	299 (78.3%)	0.015
Ivabradine	23 (5.5%)	30 (4.0%)	18 (4.7%)	0.495
Furosemide	393 (92.7%)	677 (90.6%)	361 (90.5%)	0.421
Aspirin	185 (44.2%)	318 (42.5%)	183 (48.3%)	0.183
Thienopyridines	64 (15.4%)	161 (21.6%)	89 (23.5%)	<0.001
Novel oral anticoagulants	154 (24.8%)	262 (28.6%)	113 (18.7%)	<0.001
Warfarin	167 (26.9%)	221 (24.2%)	148 (24.6%)	0.455
Intravenous inotropes	11 (2.6%)	3 (0.4%)	6 (1.6%)	0.003

* Descriptive statistics are based on count (%) or median (1st quartile; 3rd quartile), whereas *p* values for inferential statistics are based on Fisher exact or Kruskal–Wallis tests (with ^¶^, ^#^, and ^$^ representing, respectively, *p* < 0.05 at Dunn post hoc tests for tertile 1 vs. tertile 2, tertile 1 vs. tertile 3, and tertile 2 vs. tertile 3).

**Table 5 medicina-61-00904-t005:** Unadjusted and adjusted survival analysis according to tertiles of monthly volume, center-wise *.

Outcome	Unadjusted Effect Estimates	Adjusted Effect Estimates
Death		
Tertile 2 vs. 1	0.78 (0.64–0.96), *p* = 0.018	0.60 (0.19–1.82), *p* = 0.364
Tertile 3 vs. 1	0.90 (0.71–1.13), *p* = 0.365	0.54 (0.16–1.78), *p* = 0.308
Tertile 3 vs. 2	1.28 (0.93–1.42), *p* = 0.191	0.90 (0.36–2.21), *p* = 0.814
Cardiac death		
Tertile 2 vs. 1	0.70 (0.53–0.93), *p* = 0.015	0.55 (0.14–2.21), *p* = 0.398
Tertile 3 vs. 1	0.89 (0.65–1.21), *p* = 0.456	0.44 (0.09–2.12), *p* = 0.304
Tertile 3 vs. 2	1.26 (0.94–1.69), *p* = 0.117	0.80 (0.24–2.68), *p* = 0.713
Death or hospitalization		
Tertile 2 vs. 1	0.80 (0.67–0.96), *p* = 0.014	0.63 (0.24–1.66), *p* = 0.351
Tertile 3 vs. 1	0.89 (0.72–1.09), *p* = 0.246	0.37 (0.13–1.07), *p* = 0.065
Tertile 3 vs. 2	1.11 (0.92–1.34), *p* = 0.265	0.59 (0.25–1.36), *p* = 0.213
Cardiac death or hospitalization for heart failure		
Tertile 2 vs. 1	0.64 (0.50–0.82), *p* < 0.001	0.62 (0.16–2.37), *p* = 0.488
Tertile 3 vs. 1	0.83 (0.64–1.09), *p* < 0.001	0.16 (0.03–0.81), *p* = 0.026
Tertile 3 vs. 2	1.30 (1.01–1.69), *p* = 0.045	0.26 (0.07–0.99), *p* = 0.048

* Reported as hazard ratio (95% confidence interval), *p* value.

## Data Availability

The datasets presented in this article are not readily available because of inclusion of patient details. Requests to access the datasets should be directed to the corresponding author.

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
