# Peer review of "Impact of Institutional Monthly Volume of Transcatheter Edge-to-Edge Repair Procedures for Significant Mitral Regurgitation: Evidence from the GIOTTO-VAT Study"

_medicina, 2025, doi:10.3390/medicina61050904_

Round 1
Reviewer 1 Report
Comments and Suggestions for Authors
I reviewed with interest the manuscript by Nicola Corcione et al. "Impact of institutional monthly volume of transcatheter edge-to-edge repair procedures for significant mitral regurgitation: evidence from the GIOTTO-VAT study". In this article, the authors present the results of a retrospective analysis of data obtained from the GIOTTO registry, depending on the volume of procedures in individual centers. The results of the study confirmed the authors' initial hypothesis. High monthly institutional volume of TEER mitral valve repair appears to correlate with improved procedural success rates and shorter hospital stays. Similar favorable results were obtained for long-term rates of cardiac death or hospitalization for heart failure. The known facts about the success of surgical procedures with high center experience were confirmed for the transcatheter edge-to-edge repair procedure. While reviewing, I had the following comments and questions:
1. The text of the manuscript does not clearly formulate the purpose of the study (unlike ABSTRACT). I think this section of the Introduction needs to be improved.
2. In the list of references, 8 out of 24 sources are publications of the authors of the article. In my opinion, this exceeds the acceptable threshold of self-citation. Haven't other scientists studied the problem of transcatheter edge-to-edge repair procedures?
3. The Discussion section does not include a section on Limitations of the study.
4. I think it would be appropriate to consider the following publications on the topic in the article - refs. 1-2, see below.
References:
1. Steitieh D, Zaidi A, Xu S, Cheung JW, Feldman DN, Reisman M, Mallya S, Paul TK, Singh HS, Bergman G, Vadaketh K, Naguib M, Minutello RM, Wong SC, Amin NP, Kim LK. Racial Disparities in Access to High-Volume Mitral Valve Transcatheter Edge-to-Edge Repair Centers. J Soc Cardiovasc Angiogr Interv. 2022 Jul 13;1(5):100398. doi: 10.1016/j.jscai.2022.100398.
2. Grayburn PA, Mack MJ, Manandhar P, Kosinski AS, Sannino A, Smith RL 2nd, Szerlip M, Vemulapalli S. Comparison of Transcatheter Edge-to-Edge Mitral Valve Repair for Primary Mitral Regurgitation Outcomes to Hospital Volumes of Surgical Mitral Valve Repair. Circ Cardiovasc Interv. 2024 Apr;17(4):e013581. doi: 10.1161/CIRCINTERVENTIONS.123.013581.
Author Response
I reviewed with interest the manuscript by Nicola Corcione et al. "Impact of institutional monthly volume of transcatheter edge-to-edge repair procedures for significant mitral regurgitation: evidence from the GIOTTO-VAT study". In this article, the authors present the results of a retrospective analysis of data obtained from the GIOTTO registry, depending on the volume of procedures in individual centers. The results of the study confirmed the authors' initial hypothesis. High monthly institutional volume of TEER mitral valve repair appears to correlate with improved procedural success rates and shorter hospital stays. Similar favorable results were obtained for long-term rates of cardiac death or hospitalization for heart failure. The known facts about the success of surgical procedures with high center experience were confirmed for the transcatheter edge-to-edge repair procedure.
Reply: We thank the reviewer for the careful review of our work.
While reviewing, I had the following comments and questions:
1. The text of the manuscript does not clearly formulate the purpose of the study (unlike ABSTRACT). I think this section of the Introduction needs to be improved.
Reply: We have edited the Introduction section to explicitly state the hypothesis and aim of our analysis.
In the list of references, 8 out of 24 sources are publications of the authors of the article. In my opinion, this exceeds the acceptable threshold of self-citation. Haven't other scientists studied the problem of transcatheter edge-to-edge repair procedures?
Reply: We thank the reviewer for pointing out this issue. We indeed opted to quote extensively prior reports from the GIOTTO registry to establish its methodological strengths and multidimensional insights. We understand however that it is crucial to limit self-citations, in both relative and absolute terms.
The Discussion section does not include a section on Limitations of the study.
Reply: We have expanded the Discussion section to openly acknowledge the limitations of our work.
I think it would be appropriate to consider the following publications on the topic in the article - refs. 1-2, see below.
References:
1. Steitieh D, Zaidi A, Xu S, Cheung JW, Feldman DN, Reisman M, Mallya S, Paul TK, Singh HS, Bergman G, Vadaketh K, Naguib M, Minutello RM, Wong SC, Amin NP, Kim LK. Racial Disparities in Access to High-Volume Mitral Valve Transcatheter Edge-to-Edge Repair Centers. J Soc Cardiovasc Angiogr Interv. 2022 Jul 13;1(5):100398. doi: 10.1016/j.jscai.2022.100398.
2. Grayburn PA, Mack MJ, Manandhar P, Kosinski AS, Sannino A, Smith RL 2nd, Szerlip M, Vemulapalli S. Comparison of Transcatheter Edge-to-Edge Mitral Valve Repair for Primary Mitral Regurgitation Outcomes to Hospital Volumes of Surgical Mitral Valve Repair. Circ Cardiovasc Interv. 2024 Apr;17(4):e013581. doi: 10.1161/CIRCINTERVENTIONS.123.013581.
Reply: We thank the reviewer for pointing out these important works, which are now openly quoted in the Discussion.
Reviewer 2 Report
Comments and Suggestions for Authors
The authors present their article titled "Impact of institutional monthly volume of transcatheter edge-to-edge repair procedures for significant mitral regurgitation: evidence from the GIOTTO-VAT study" where they study the association between institutional volume and outcomes of TEER procedures.
They report some differences on patients' baseline characteristics suggesting higher risk cases in high volume centres while left ventricular function being somewhat more common in low volume centres. High volume centres had higher rates of procedural success with fewer rates of residual regurgitation and bleeding events and shorter hospital stay. Finally, on an adjusted survival analysis, high volume centres had a lower rate of cardiac mortality and heart failure hospitalizations.
The article is very well written with a very interesting concept and proper statistical analysis for the most part. However there are a couple of issues that need to be addressed:
Major issues
1. I suggest that Kruskal-Wallis test to be followed-up with a post-hoc test with the low volume centres as a reference point for a more complete analysis wherever the Kruskal-Wallis test is statistically significant.
2. It would be beneficial to list the confounders used for the adjusted survival analysis.
Author Response
Reviewer #2
The authors present their article titled "Impact of institutional monthly volume of transcatheter edge-to-edge repair procedures for significant mitral regurgitation: evidence from the GIOTTO-VAT study" where they study the association between institutional volume and outcomes of TEER procedures.
They report some differences on patients' baseline characteristics suggesting higher risk cases in high volume centres while left ventricular function being somewhat more common in low volume centres. High volume centres had higher rates of procedural success with fewer rates of residual regurgitation and bleeding events and shorter hospital stay. Finally, on an adjusted survival analysis, high volume centres had a lower rate of cardiac mortality and heart failure hospitalizations.
Reply: We thank the reviewer for the supportive review of our work.
The article is very well written with a very interesting concept and proper statistical analysis for the most part. However there are a couple of issues that need to be addressed:
Major issues
- I suggest that Kruskal-Wallis test to be followed-up with a post-hoc test with the low volume centres as a reference point for a more complete analysis wherever the Kruskal-Wallis test is statistically significant.
Reply: As requested, we have added results of Dunn post-hoc tests to Tables 1-4.
- It would be beneficial to list the confounders used for the adjusted survival analysis.
Reply: We thank the reviewer for raising this important methodological aspect. Indeed, the following potential confounders were forced into the adjusted models: age, gender, smoking history, dyslipidemia, degenerative etiology, baseline functional class, prior mitral valve repair, prior stroke, frailty, peripheral artery disease, surgical risk scores, left atrial diameters, left ventricular diameters, left ventricular volumes, tenting area, severe calcification, prolapse, severity of tricuspid regurgitation, concomitant ECG abnormalities, atrial fibrillation, and significant coronary artery disease. This detail has also been added to the Methods section.
Round 2
Reviewer 1 Report
Comments and Suggestions for Authors
The authors responded to my comments and made changes to the text of the manuscript. However, I still have comments on the article. 1. The authors claim in their response to the 1st comment that they formulated the purpose of the study in the Introduction section. However, I again did not see this formulation.
Author Response
Reply: We have revised further the Introduction, adding the following phrase: ‘We thus aimed at appraising the impact of institutional monthly volume of TEER on early and long-term outcomes’.
Reviewer 2 Report
Comments and Suggestions for Authors
The authors have adequately responded to both of my comments. In particular:
- They have added the post-hoc results of the Kruskal-Wallis test in Tables 1-4 using the Dunn's test.
- They have listed the potential confounders that were used in the adjusted models in the Methods section.
Author Response
Reply: We thank the reviewer for the supportive appraisal of our revised manuscript.